# Cytokine Profiles Differentiate Symptomatic from Asymptomatic PTSD in Service Members and Veterans with Chronic Traumatic Brain Injury

**DOI:** 10.3390/biomedicines10123289

**Published:** 2022-12-19

**Authors:** Ethan G. Smith, James Hentig, Carina Martin, Chelsea Wagner, Vivian A. Guedes, Katie A. Edwards, Christina Devoto, Kerri Dunbar, Michael J. Roy, Jessica M. Gill

**Affiliations:** 1National Institute of Nursing Research, National Institutes of Health, Bethesda, MD 20892, USA; 2Traumatic Brain Injury Center of Excellence (TBI CoE), Research and Development Directorate (J-9), Defense Health Agency, Fort Carson, CO 80913, USA; 3Intrepid Spirit Center, Evans Army Community Hospital, Fort Carson, CO 80913, USA; 4General Dynamics Information Technology, Falls Church, VA 22042, USA; 5Center for Neuroscience and Regenerative Medicine, Uniformed Services University of the Health Sciences, Bethesda, MD 20814, USA; 6School of Nursing and Medicine, Johns Hopkins University, Baltimore, MD 21205, USA; 7Department of Medicine, Uniformed Services University of Health Sciences, Bethesda, MD 20814, USA; 8The Henry M. Jackson Foundation for the Advancement of Military Medicine, Bethesda, MD 20817, USA

**Keywords:** military, TBI, cytokines, IL-8, PTSD, inflammation

## Abstract

Traumatic brain injuries (TBI) and posttraumatic stress disorder (PTSD) are commonly observed comorbid occurrences among military service members and veterans (SMVs). In this cross-sectional study, SMVs with a history of TBI were stratified into symptomatic and asymptomatic PTSD groups based on posttraumatic stress checklist-civilian (PCL-C) total scores. Blood-based biomarkers were assessed, and significant differential markers were associated with scores from multiple neurobehavioral self-report assessments. PCL-C cutoffs were total scores >50 (PTSD symptomatic) and <25 (asymptomatic). Cytokines IL6, IL8, TNFα, and IL10 were significantly elevated (*p* < 0.05–0.001) in the TBI+/PTSD symptomatic group compared to the TBI+/asymptomatic group. Cytokine levels of IL8, TNFα, and IL10 were strongly associated with PCL-C scores (0.356 < r > 0.624 for all, *p* < 0.01 for all), while TNFα and IL10 were additionally associated with NSI totals (r = 0.285 and r = 0.270, *p* < 0.05, respectively). This is the first study focused on PTSD symptom severity to report levels of circulating pro-inflammatory IL8, specifically in SMVs with TBI. These data suggest that within the military TBI population, there are unique cytokine profiles that relate to neurobehavioral outcomes associated with TBI and PTSD.

## 1. Introduction

Traumatic brain injuries (TBI) have become a major concern of the military medical and research communities following their increased prevalence among service members and veterans (SMVs) during military training and combat operations [1,2]. From 2000 to 2021, approximately 450,000 U.S. service members sustained a TBI, 82.3% of which were determined to be mild TBI (mTBI) [3]. Having a history of TBI is associated with a higher risk for multiple psychiatric disorders, and just a single incidence of mTBI has been found to increase a patient’s risk of developing dementia. Additionally, increased incidences of depression, anxiety, and suicidal ideation have been reported following TBI [4,5,6], placing SMVs, who have a high occupational risk of TBI, also at higher risk of developing associated psychiatric sequalae.

TBI and post-traumatic stress disorder (PTSD), along with pain, make up the polytrauma triad and have been studied extensively. PTSD is a complex and multidimensional disorder that includes various subtypes with different neurobiological underpinnings (intrusive thoughts, avoidance, anger, substance abuse, and isolation) and has a strong relationship with suicide risk [7,8] In a review of 33 publications reporting on TBI and PTSD in U.S. SMVs, Greer [5] found that having a history of mTBI was linked to the prevalence of PTSD. When accounting for the potential confounding effects of pre-deployment mental health symptoms, prior TBI, and combat intensity, multiple studies found that sustaining a TBI during deployment was the greatest predictor for the development of PTSD within a year post-deployment [9,10]. Although there are factors unique to the military experience that likely influence the development of PTSD, similar associations between TBI and PTSD have been observed in civilian populations [11,12,13], suggesting TBI may play a role in the development of PTSD. Importantly, disorders such as PTSD can arise in the chronic phase of TBI, years after the initial injury [13]. Additionally, several studies, including those with SMV cohorts, have reported that sustaining a mTBI, has an association with an enhanced risk for PTSD development [11,12,13]. Thus, additional tools are necessary to identify SMVs whose TBI indicates they are at greater risk for subsequent psychiatric sequelae.

TBI can significantly disrupt the integrity of the blood–brain barrier, resulting in the potential for infiltration of macrophages and the exchange of neuroinflammatory markers being distributed throughout the peripheral blood [14]. Following injury, a molecular cascade activates resident immune cells, which polarize towards inflammatory and anti-inflammatory activities, thereby releasing associated cytokines. Inflammatory cytokines include interleukin 1 (IL1), interleukin 6 (IL6), interleukin 8 (IL8) and tumor necrosis factor alpha (TNFα), among others. Anti-inflammatory cytokines include interleukin 10 (IL10), which triggers the release of interleukin 1 receptor antagonist (IL1RA), preventing inflammatory IL1 signaling through competitive inhibition [14]. Peripheral blood biomarkers of inflammation have been associated with both TBI and PTSD; specifically, elevated IL6, TNFα, and IL10 have been strongly associated with repetitive TBI and increased PTSD symptom severity [15]. The isolation of blood-based biomarkers by assessment of inflammatory proteins within neuronal-derived extracellular vesicles (EV) from SMVs with and without mTBI has confirmed elevated levels of EV IL10 and TNFα in SMVs with a history of mTBI and their association with PTSD and depression symptoms [16]. Similarly, a history of close blast exposure was reported with elevated levels of IL6 and TNFα, which were in turn associated with greater deployment-related psychiatric sequelae, including greater PTSD symptom severity [17]. IL8 has been associated with decreased depression in females and non-significant results in males within civilian non-TBI samples [18]; however, other studies have found that SMVs who met the criteria for depression 12 months after sustaining a TBI had higher levels of IL8 compared to SMVs who had sustained TBI but did not meet criteria for depression [19]. To the best of our knowledge, no study has assessed the relationship between IL8 and PTSD in a cohort of SMVs.

The current state of evidence suggests a strong link between TBI, PTSD, and inflammation; however, few studies have assessed populations with chronic symptoms beyond 2 years post-injury [15,17]. Here, we aimed to assess whether there were differences in the peripheral blood concentrations of CRP, IL1RA, IL6, IL8, IL10, and TNFα in a cohort of SMVs with a history of TBI, with a longer average time since injury (TSI), who were either symptomatic or asymptomatic for PTSD. Although some of these cytokines have been previously reported, CRP, IL1RA, and IL8 have not been studied specifically in the context of SMVs with TBI, nor have they been linked to PTSD symptom severity. Furthermore, we report how these cytokines correlate with measures of behavioral symptoms, including PTSD, depression, and post-concussive syndrome (PCS) symptoms, as well as health-related quality of life (HRQOL). Study findings provide a framework for the clinical utility of identifying individuals who are at risk of chronic psychiatric sequelae following TBI, and links to HRQOL.

## 2. Materials and Methods

### 2.1. Study Design

Research participants (*n* = 513) were enrolled (*n* = 123 analyzed) as part of a larger ongoing study at the Walter Reed National Military Medical Center (WRNMMC, Bethesda, MD, USA) or Fort Belvoir Community Hospital (FBCH, Fort Belvoir, VA, USA). Institutional review boards at WRNMMC, FBCH and the Uniformed Services University of the Health Sciences (USUHS) approved this study. Informed consent was obtained from all subjects involved in the study. After acquiring written witnessed informed consent, participants underwent a medical history and physical exam and had blood samples drawn. In addition, participants completed a battery of self-report questionnaires including the PTSD Checklist—Civilian Version (PCL-C), Combat Exposure Scale (CES), Satisfaction with Life Scale (SWLS), Epworth Sleepiness Scale (ESS), Neurobehavioral Symptom Inventory (NSI), Patient Health Questionnairre-9 (PHQ9), and HRQOL Short Form Survey (SF-36). Participants without a lifetime history of TBI (*n* = 95), missing blood draw (*n* = 56), incomplete demographics (*n* = 10), over the age of 55 (*n* = 54), SMVs with a PCL-C score between thresholds for symptomatic and asymptomatic (*n* = 135), as well as a subpopulation of civilians with no prior military service (*n* = 40) included in the larger ongoing study were excluded from this analysis.

### 2.2. Traumatic Brain Injury Characterization

A lifetime history of TBI was characterized using the Ohio State University Traumatic Brain Injury Identification Method (OSU TBI-ID), a structured interview carried out by trained research staff. A positive TBI was defined as a head insult that resulted in loss of consciousness (LOC) and/or alterations of consciousness (AOC), and TBI severity was classified using self-reported LOC and AOC according to Veterans Affairs/Department of Defense guidelines (The Management of Concussion-Mild TBI Working Group, 2009). Some participants experienced multiple TBIs in their lifetime. Thus, the most recent and the most severe TBI (highest severity by LOC) were used to classify TBI severity. Time since the last injury (TSI) was defined as the difference between the date of blood collection and the date of the most recent TBI.

### 2.3. Participant Grouping Based on PCL-C

To assess the effects of PTSD symptom severity on plasma cytokine levels, participants were stratified into either a PTSD symptomatic group or a control asymptomatic group. Groups were based on PTSD symptom severity as measured by the PTSD Checklist—Civilian Version (PCL-C), a 17-item questionnaire that scores each question on a scale of 1 (not at all) to 5 (extremely). The PCL-C total score ranges from 17 to 85 with higher scores indicating greater symptom severity. The PCL-C has three symptom clusters based on DSM-IV characterization of PTSD symptomology: symptoms of reexperiencing, avoidance and numbing, and arousal, each with its own subscale score. This assessment has been shown to have excellent internal consistency (α = 0.94), and test-retest coefficient values range from 0.64 to 0.92. The PCL-C was also shown to have acceptable convergent validity (r > 0.75) [20].

The PTSD symptomatic group consisted of SMVs with PCL-C scores above 50 (*n* = 62), while the control asymptomatic group consisted of SMVs with PCL-C scores below 25 (*n* = 61). Any SMVs with PCL-C scores between 25 and 50 were excluded from the analysis.

### 2.4. Cytokine Concentration Measurement

Following standard laboratory procedures, plasma was isolated from blood and stored at −80 °C until batch processing. Plasma protein concentration was measured using the MESO QuickPlex SQ 120 (MesoScale Diagnostics [MSD], Rockville, MD, USA). On the day of analyte measurement, plasma samples were thawed on ice, and plasma cytokine concentrations were measured using three V-PLEX assays according to the manufacturer’s instructions: one for IL1RA, one for CRP, and one for IL6, IL8, IL10, and TNFα. Plasma samples were run in duplicate and averaged. The lower limit of quantification for each protein was 9.19, 27.6, 0.633, 0.591, 0.298, and 0.690 for IL1RA, CRP, IL6, IL8, IL10, and TNFα, respectively. Protein measurements with CVs above 20% were excluded from their respective analysis. In addition, any protein measurements +/− 3 standard deviations from the mean were considered outliers and thus excluded from the analysis.

### 2.5. Self-Report Behavioral Symptom Measures

Combat exposure was assessed using the Combat Exposure Scale (CES), a seven-item questionnaire that assesses the degree to which SMVs experience wartime stressors. Each item is rated on a scale of 1 to 5. The CES total score ranges from 0 to 41 with higher scores indicating greater subjective combat exposure. The CES possesses good internal consistency (α = 0.85) and excellent test-retest reliability (r = 0.97) [21]. 

Satisfaction with life was measured using the Satisfaction with Life Scale (SWLS), a five-item questionnaire that assesses a person’s global life satisfaction. Each item has a 7-point Likert scale ranging from “Strongly Disagree” to “Strongly Agree”. The SWLS total score ranges from 5 (extremely dissatisfied) to 35 (extremely satisfied), and the instrument has good internal consistency (α = 0.87) and test-retest reliability (r = 0.820) [22]

Daytime sleepiness was measured using the Epworth Sleepiness Scale (ESS), an eight-item questionnaire with each item asking the participant, in different daytime situations, to rate the likelihood of dozing on a scale of 0 being “no chance of dozing” to 3 being “high chance of dozing”. The ESS total score ranges from 0 to 24 with greater scores indicating greater daytime sleepiness, and the instrument has been shown to have good internal consistency (α = 0.88) and test-retest reliability (r = 0.82) [23].

Post-concussive symptoms were measured using the Neurobehavioral Symptom Inventory (NSI), a 22-item questionnaire with each item rated on a scale of 0 (none) to 4 (severe). The total score ranges from 0 to 88 with higher scores indicating worse symptoms and was further examined by subscale scores: vestibular, somatosensory, cognitive, and affective. For the NSI total score, Cronbach’s alpha and test-retest reliability values are estimated to range from 0.81 to 0.96 and 0.74 to 0.94 [24], respectively. Cronbach’s alpha exceeds 0.80 for each NSI subscale, while test-retest reliability values range from 0.52 to 0.91.

The severity of depression symptoms was assessed using the Patient Health Questionnairre-9 (PHQ9), a nine-item questionnaire with each item asking the participant to rate, on a scale of 0 (not at all) to 3 (nearly every day), how often they were bothered by the problem presented in each item. An example item includes “Little interest or pleasure in doing things”. The total score ranges from 0 to 27 with higher scores denoting greater symptom severity. The PHQ9 has excellent sensitivity (0.93) and specificity (0.89) for detecting major depression disorder in a population of TBI-inflicted adults, and the test-rest reliability is acceptable (r = 0.76 and κ = 0.460) [25].

HRQOL was assessed using the 36-item Short Form Survey (SF-36). The questionnaire measures both physical and mental HRQOL and is separated into eight subscales: physical functioning, role limitations due to physical problems, role limitations due to emotional problems, vitality, emotional well-being, social functioning, pain, and general health. Each subscale score ranges from 0 to 100 with higher scores indicating better HRQOL. Internal consistency and test-rest coefficient estimates of reliability consistently exceed 0.70 across multiple studies validating the SF-36. In addition, the validity of the SF-36 is acceptable (r ≥ 0.4) [26].

### 2.6. Statistical Methods

Statistical analyses were conducted using SPSS GradPack 24.0. Figures were created using GraphPad Prism 9 (version 9.0.0). Through an analysis of histogram distribution and skewness and kurtosis statistics, continuous demographic, clinical, and cytokine variables were determined to be non-normally distributed. Thus, the non-parametric Mann–Whitney U test was used to assess for statistically significant differences between the PTSD symptomatic and control asymptomatic groups. To assess group differences for categorical variables, either Fisher’s exact test or Pearson’s chi-squared test was used. Binomial logistic regression models were built to control for variables believed to influence cytokine concentration results. TBI severity was excluded from the model because the variable was skewed, with most participants reporting mTBI. In addition, the NSI total was excluded from the model as it was determined to be colinear with the cytokine measures. The variables included in the final model were BMI, number of TBIs, TSI, and CES total. The cytokine concentration data were log base 2 transformed to decrease the skewness of the distribution thereby improving the fit of the data in the binomial logistic regression model. A binomial logistic regression model was created for each cytokine found to be significant between groups with the covariates mentioned above to determine if cytokine differences maintained statistical significance. Lastly, we conducted Spearmen’s correlation analysis between the cytokine data and self-report measures of behavioral symptoms. A simple linear regression line was fit to each figure representing correlational data.

## 3. Results

### 3.1. Demographic and Clinical Characteristics

The study cohort included 62 participants in the PTSD symptomatic group and 61 participants in the control asymptomatic group; the two groups were similar across a majority of demographic and clinical variables (Table 1). Both groups were predominately male, white, and active-duty military. The significantly different variables between groups were BMI, lifetime number of TBIs, TSI, and most severe TBI over a lifetime. The PTSD symptomatic group had a greater BMI (median = 28.75, IQR: 26.30–32.13) compared to the control asymptomatic group (median = 27.30, IQR: 24.65–29.85, *p* = 0.031). The PTSD symptomatic group reported a greater number of TBIs (median = 4.00, IQR: 2.00–6.00) and shorter TSI (median = 4.26, IQR: 1.54–8.70) compared to the control asymptomatic group (# TBI: median = 2.00, IQR: 1.00–4.00, *p* < 0.001; TSI: median = 8.26, IQR: 3.31–14.13, *p* = 0.002). Additionally, there were significant differences across groups when comparing TBI severity (*p* = 0.034), although, a majority of participants in both groups reported mTBI as the most recent TBI and the most severe TBI. 

### 3.2. Behavioral Symptoms

Groups were stratified by PCL-C total scores with participants in the PTSD symptomatic group reporting significantly higher scores (PCL-C > 50) compared to the control asymptomatic group (PCL-C < 25), as well as higher scores for each PCL-C subscale: reexperiencing, avoidance/numbing, and arousal (*p* < 0.001, Table 1). The PTSD symptomatic group reported significantly higher NSI scores, as well as higher scores for each of the four NSI subscales (*p* < 0.001, Table 1). Additionally, we assessed HRQOL, in which the PTSD symptomatic group had significantly lower scores for the SF-36 subscales of physical functioning (*p* < 0.05) and social functioning (*p* < 0.01) compared to the control asymptomatic group (Table 1). The other six SF-36 subscales were similar across groups (Table 1). Although the PTSD symptomatic group had higher scores of CES total (<0.001), there were no significant differences for the total scores of PHQ9, SWLS, or ESS (Table 1).

### 3.3. Protein Biomarkers

Pairwise comparisons of plasma levels of cytokines were assessed and the PTSD symptomatic group had significantly elevated levels of both pro- and anti-inflammatory cytokines. Proinflammatory cytokines IL6 (median = 0.2063, IQR: 0.1494–0.2982), IL8 (median = 2.4104, IQR: 1.6175–3.2842), and TNFα (median = 0.7836, IQR: 0.3540–1.0373) were significantly elevated compared to the control asymptomatic group (IL6: median = 0.1343, IQR: 0.0949–0.2268, *d* = 0.5; IL8: median = 1.5922, IQR: 1.2352–2.1294, *d* = 0.84; TNFα: median = 0.2652, IQR: 0.2234–0.3584, *d* = 1.34, all *p* < 0.001, Figure 1A–C). Additionally, anti-inflammatory IL10 was also significantly higher (*p* = 0.026, *d* = 0.55) in the PTSD symptomatic group (median = 0.1200, IQR: 0.0841–0.1813) compared to the asymptomatic group (median = 0.0963, IQR: 0.0615–0.1335, *p* < 0.05, Figure 1D). No significant differences were found in CRP (*p* = 0.093), IL1RA (*p* = 0.427), VEGF (*p* = 0.614), or ptau (*p* = 0.398) between groups ( Appendix A). After controlling for BMI, number of TBIs, TSI, and CES total scores, in logistic regression analyses, IL6, IL8, TNFα, and IL10 maintained significance between the PTSD symptomatic and control asymptomatic groups (Table 2).

### 3.4. Correlation Analysis between Plasma Proteins and Behavioral Symptoms

Plasma cytokine levels strongly correlated with patient-reported neurobehavioral measures in the PTSD symptomatic group. Cytokine levels of IL8, TNFα, and IL10 are strongly associated with PCL-C (0.356 < r > 0.624 for all, *p* < 0.01 for all, Figure 2). With IL8, we also observed significant positive correlations with PCL-C reexperiencing (r = 0.325, *p* < 0.05) and PCL-C avoidance/numbing (r = 0.276, *p* < 0.05, Figure 2). We observed that IL10 was also positively associated with NSI total (r = 0.270, *p* < 0.05) and NSI affective subfactor (r = 0.268, *p* < 0.05), while inversely correlated with SF36 social functioning (r = −0.295, *p* < 0.05, Figure 2). Finally, in our correlation analysis, we observed the highest number of significant correlations with TNFα, which was positively associated with multiple PCL-C subfactors, PCL-C reexperiencing, PCL-C avoidance/numbing, and PCL-C arousal (0.316 < r > 0.464, for all, *p* < 0.05–0.01 for all), as well as with NSI total (r = 0.285, *p* < 0.05, Figure 2).

## 4. Discussion

In the present study, we indicate that SMVs with chronic TBI differ based on cytokine profiles when comparing those with high and low PTSD symptoms and that elevations in these inflammatory cytokines correlate with adverse neurobehavioral symptoms and perceptions of health quality. Specifically, we report that in PTSD-symptomatic SMVs, traditional cytokine biomarkers IL6, IL10, and TNFα were elevated, along with IL8, and were associated with deteriorating neurobehavioral outcomes. To our knowledge, this study is the first to report levels of circulating pro-inflammatory IL8 specifically in SMVs with TBI and compared using PTSD symptom severity in a cohort remote from injury.

In this cross-sectional analysis, we observed elevated levels of IL8, an interleukin and chemoattractant cytokine that activates neutrophils following a range of inflammatory assaults including CNS injury [27,28]. IL8 is known to be synthesized and released following stimulation by TNFα [29,30] and has been associated with both TBI [30,31] and PTSD [32,33]. Military-associated TBI and PTSD both can result in unwanted intrusive thoughts [34,35]. We observed that both elevated TNFα and IL8 were associated with PCL-C subcategories of reexperiencing and avoidance/numbing, suggesting these inflammatory cytokines may be a contributing factor. However, in prior explorations of TBI and PTSD, IL8 has yielded mixed results [18,19,32]. These contrasting findings could be the result of covariates such as the mechanism of injury (blunt vs. blast vs. penetrating TBI), time since injury in TBI studies, and unreported alcohol or substance use, personality disorders, or anxiety in PTSD investigations.

The directionality of IL8 in relation to neurobehavioral outcomes is interesting. Multiple studies have reported decreased IL8 associated with treatment-resistant depression [18], anxiety [36], and suicidality [37]; however, these studies were conducted in non-TBI civilian populations. In contrast, Chaban et al. [38] reported elevated IL8 in a civilian TBI population although they did not examine neurobehavioral outcomes, and consistent with our findings, Juengst et al. [19] reported an increase in IL8 expression in a military TBI population that was associated with post-traumatic depression. The discernable difference is the presence of a TBI. However, several studies have reported sex differences in biomarkers in both TBI and/or neurobehavioral studies [18,39], with female participants notably low in many military studies.

Although our report on IL8 adds to the confounding conclusions of the cytokine in both TBI and PTSD literature, our findings of elevated IL6, TNFα, and IL10 are in concordance with previous reports. It has been repeatedly demonstrated that plasma levels of IL6 and TNFα were significantly elevated in patients with TBI compared to uninjured controls [40]. Lima [41] observed increased IL6 in patients with mental stress, which was further increased in patients with PTSD, and Hussein [42] performed a meta-analysis of 27 cross-sectional studies that documented elevated TNFα in those with PTSD.

Our findings have some limitations. First, while our analyzed sample is ~18% female, mirroring that of the U.S. active duty population, larger numbers of women in particular would allow for greater elucidation into driving factors of differential outcomes observed in the literature. Similarly, a better-matched cohort in terms of TSI, number of lifetime TBIs, and most severe injury may lead to different future findings. Finally, although PHQ9 is a robust and routinely used tool, others exist that permit the analysis of subscales or single items and are clinician-rated. Thus, it is important to determine whether our findings can be replicated in other populations, ideally with greater numbers. Despite these limitations, our demonstration of associations between PTSD, inflammatory biomarkers, and perceived health provides further evidence of interactions among health and psychiatric symptoms following a TBI and suggests the need to intervene to prevent the impacts of sustained inflammation over time.

## Figures and Tables

**Figure 1 biomedicines-10-03289-f001:**
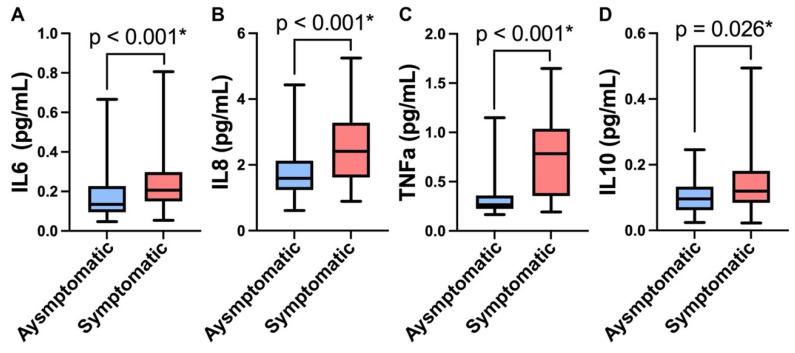
**Group differences in plasma cytokine concentration.** The cytokine concentration distribution is shown for the PTSD symptomatic and asymptomatic control groups using boxplots indicating the median, IQR, and minimum and maximum values. Cytokines IL6 (**A**), IL8 (**B**), TNFα (**C**), and IL10 (**D**) were all elevated in the PTSD symptomatic group. The Mann-Whitney U test was used to assess group differences. *p* values are reported above each plot (* indicates statistical significance at the 0.05 level).

**Figure 2 biomedicines-10-03289-f002:**
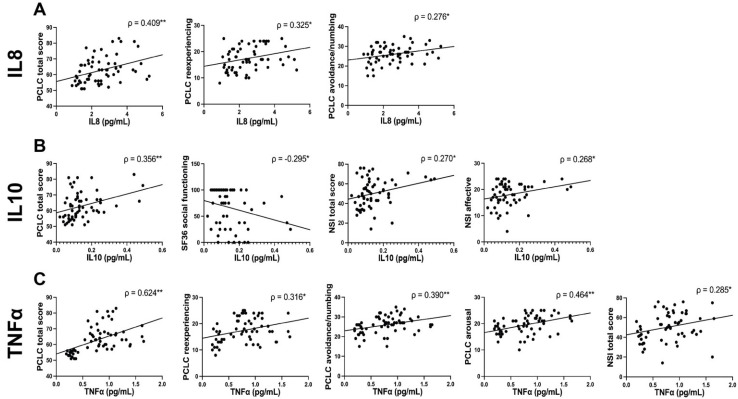
**Scatter plots of significant correlations in the PTSD high group.** In the PTSD symptomatic group, significant correlations between cytokine concentrations and behavioral measures are represented via scatter plots. Spearman’s rho correlation coefficient (ρ) was calculated as the data were non-normal. The Spearman’s rho correlation coefficient is represented on each scatter plot with an asterisk indicating the significance level (*p* * < 0.05, *p* ** < 0.01).

**Table 1 biomedicines-10-03289-t001:** **Demographic and clinical characteristics.**

Characteristic	PTSD High, N = 62 ^1^	PTSD Low, N = 61 ^1^	*p*-Value ^2^	Cohen’s d	95% Confidence Interval
Age	37.00 (31.00, 45.00)	36.00 (29.50, 43.50)	0.474	−0.16	(−0.51, 0.20)
Body Mass Index (kg/m^2^)	28.75 (26.30, 32.13)	27.30 (24.65, 29.85)	0.031 *	−0.39	(−0.75, −0.04)
Sex			0.165		
Male	47 (75.8%)	53 (86.9%)			
Female	15 (24.2%)	8 (13.1%)			
Race			0.529		
White	44 (71%)	50 (82%)			
Black or African-American	13 (21%)	7 (11.5%)			
Asian	3 (4.8%)	3 (4.9%)			
American Indian or Alaska Native/Inuit	1 (1.6%)	1 (1.6%)			
Native Hawaiian or Other Pacific Islander	1 (1.6%)	0 (0%)			
Ethnicity			0.603		
Not Hispanic or Latino	52 (83.9%)	54 (88.5%)			
Hispanic or Latino	10 (16.1%)	7 (11.5%)			
Military Status			0.108		
Active duty military	42 (67.7%)	51 (83.6%)			
Retired from military	10 (16.1%)	5 (8.2%)			
Veteran	7 (11.3%)	1 (1.6%)			
National Guard	1 (1.6%)	3 (4.9%)			
Reserve component	1 (1.6%)	1 (1.6%)			
Inactive reserve	1 (1.6%)	0 (0%)			
Most recent TBI severity			0.365		
mTBI	57 (91.93%)	54 (88.52%)			
moTBI	4 (6.45%)	3 (4.91%)			
sTBI	1 (1.61%)	4 (6.55%)			
Highest TBI severity			0.034 *		
mTBI	49 (79.03%)	52 (85.24%)			
moTBI	12 (19.35%)	4 (6.55%)			
sTBI	1 (1.61%)	5 (8.19%)			
Number of TBIs	4.00 (2.00, 6.00)	2.00 (1.00, 4.00)	<0.001 *	−0.67	(−1.03, −0.31)
TSI (years)	4.27 (1.54, 8.70)	8.26 (3.31, 14.13)	0.002 *	0.61	(0.25, 0.97)
CES	18.00 (8.25, 25.25)	4.00 (0.00, 17.50)	<0.001 *	−0.71	(−1.07, −0.34)
SWLS	24.00 (18.00, 29.25)	22.5 (15.25, 28.75)	0.347	−0.18	(−0.54, 0.17)
ESS	8.50 (5.00, 13.25)	10.00 (6.00, 14.00)	0.282	0.19	(−0.17, 0.54)
PHQ9	4.00 (1.00, 12.50)	12.00 (2.00, 18.00)	0.066	0.42	(0.06, 0.77)
NSI total	50.00 (41.00, 61.50)	7.00 (3.00, 13.00)	<0.001 *	−3.62	(−4.19, −3.03)
NSI vestibular	4.00 (2.00, 7.00)	0.00 (0.00, 1.00)	<0.001 *	−1.61	(−2.02, −1.20)
NSI somatosensory	12.50 (8.75, 17.00)	1.00 (0.00, 4.00)	<0.001 *	−2.20	(−2.64, −1.74)
NSI cognitive	12.00 (9.00, 14.00)	2.00 (0.00, 3.00)	<0.001 *	−2.79	(−3.29, −2.29)
NSI affective	19.00 (16.00, 21.00)	2.00 (0.50, 3.50)	<0.001 *	−4.48	(−5.15, −3.82)
PCL-C total	62.00 (56.00, 67.00)	20.00 (17.00, 22.00)	<0.001 *	−6.85	(−7.78, −5.91)
PCL-C reexperiencing	17.00 (13.75, 21.25)	5.00 (5.00, 6.00)	<0.001 *	−3.54	(−4.10, −2.96)
PCL-C avoidance numbing	26.00 (23.00, 29.00)	8.00 (7.00, 9.00)	<0.001 *	−5.59	(−6.37, −4.80)
PCL-C arousal	20.00 (17.00, 22.00)	6.00 (5.00, 7.00)	<0.001 *	−4.89	(−5.60, −4.18)
SF-36 subscales					
Physical Functioning	95.00 (78.75, 100.00)	80.00 (50.00, 100.00)	0.007 *	−0.49	(−0.85, −0.13)
Role Limitations Due To Physical Problems	50.00 (0.00, 100.00)	50.00 (0.00, 100.00)	0.829	−0.04	(−0.39, 0.32)
Role Limitations Due to Emotional Problems	66.67 (0.00, 100.00)	33.33 (0.00, 100.00)	0.061	−0.35	(−0.70, 0.01)
Vitality	47.50 (25.00, 65.00)	30.00 (12.50, 65.00)	0.063	−0.31	(−0.66, 0.05)
Emotional Well-Being	78.00 (40.00, 92.00)	60.00 (32.00, 92.00)	0.191	−0.30	(−0.66, 0.05)
Social Functioning	75.00 (37.50, 100.00)	37.50 (12.50, 100.00)	0.013 *	−0.49	(−0.85, −0.13)
Pain	62.50 (31.88, 90.00)	45.00 (22.50, 80.00)	0.162	−0.27	(−0.62, 0.09)
General Health	70.00 (40.00, 85.00)	55.00 (40.00, 80.00)	0.338	−0.17	(−0.52, 0.18)

^1^ Data are represented as median (IQR), or in the case of categorical variables, n (%). ^2^ Mann Whitney U test; Pearson chi-squared test, Fisher’s exact test, * *p* < 0.05. Abbreviations: mTBI, mild TBI; moTBI, moderate TBI; sTBI, severe TBI; TSI, time since injury; CES, combat exposure scale; SWLS, satisfaction with life scale; ESS, Epworth sleepiness scale; PHQ9, patient health questionnaire 9; NSI, neurobehavioral symptom inventory; PCL-C, PTSD checklist civilian version; SF-36, short form 36 health survey questionnaire.

**Table 2 biomedicines-10-03289-t002:** **Logistic regression of TBI + symptomatic vs. asymptomatic PTSD groups.**

	Predictors	Exp(B)	*p*-Value
Plasma IL6			
	BMI	1.040	0.451
	Number of TBIs	1.129	0.228
	Time since injury	0.928	0.024 *
	CES total	1.056	0.006 *
	Plasma IL6	2.565	0.002 *
Plasma IL8			
	BMI	1.081	0.139
	Number of TBIs	1.298	0.014 *
	Time since injury	0.951	0.115
	CES total	1.037	0.069
	Plasma IL8	4.623	<0.001 *
Plasma IL10			
	BMI	1.110	0.046 *
	Number of TBIs	1.271	0.020 *
	Time since injury	0.931	0.031 *
	CES total	1.052	0.014 *
	Plasma IL10	2.666	0.001 *
Plasma TNFα			
	BMI	1.065	0.289
	Number of TBIs	1.248	0.059
	Time since injury	0.937	0.065
	CES total	1.048	0.033 *
	Plasma TNFα	5.850	<0.001 *

* *p* < 0.05.

## Data Availability

Data available by request to appropriate and verified researchers.

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
