# Peer review of "Cytokine Profiles Differentiate Symptomatic from Asymptomatic PTSD in Service Members and Veterans with Chronic Traumatic Brain Injury"

_biomedicines, 2022, doi:10.3390/biomedicines10123289_

Round 1

Reviewer 1 Report

Thank you for the interesting study. I think there are a lot of good things in the study, but there are also unfortunately several elements that need clearing up:

Why was the PCL-C chosen? There is a military version of this checklist.

Why was the PCL-C chosen and not the current PCL-5? The PCL-C is based on DSM-IV criteria, which have been outdated for almost 10 years.

I’m even more confused, as I think you have not assessed trauma-exposure for the PCL-C, no? So, you only looked at symptoms, not trauma-exposure? Or is that done through the combat exposure scale? But, that’s odd, since the combat exposure scale would suggest that you used the PCL-M. So, what if someone has no combat exposure, but they might have had trauma-exposure in another way? In short: Has there been an selection on trauma-exposure?

When was this study performed?

Line 142: “these are suggested cutoffs for military populations and have been previously used to assess PTSD symptom severity in SMVs [21].” The reference refers to the PCL-M, not the PCL-C.

It isn’t clear is the alpha Cronbach scores refer to previous studies or the current studies. So: the alpha scores, are these for the currently used scales?

I don’t understand the population. An inactive reserve or a reserve component: do they see combat exposure? Wouldn’t they probably have not seen exposure? So, it might they don’t have PTSD, because they don’t fulfil criterion A. Which is quite important if the study is to look at relationships in explaining the role of a disorder. It might be that you are simply measuring stress symptoms.

The PHQ-9 is lower among people with PTSD than it is among people without PTSD? That’s very odd. PTSD and depression are highly correlated. The non-PTSD group has on average major depression. If this is not a mistake, I’m not sure in what way you can use these as a control group. The authors also bypass the large difference between the groups: one group has on average major depression disorder, the other does not. Even if the tests show that there is no statistical difference, it is an actual difference. Statistical test do not take into account diagnostic criteria.

I think the authors should look more into how these groups can be used as control and PTSD group. Because I don’t think they can be compared, as they are quite similar.

So, I would suggest that the authors make a better case of why they can compare these groups. Also, if it is a problem that there sample has people who have combat exposure and others who on the reserve list. Why they use the PCL-C. Et cetera.

Finally, I am unsure if the many self-references are necessary. 

Author Response

We thank the reviewers for their time and effort. Reviewing is not always an easy task, but an essential task. Thank you for your comments, suggestions, and questions that ultimately have helped shape our manuscript into a better version.

Reviewer 1:
Thank you for the interesting study. I think there are a lot of good things in the study, but there are also unfortunately several elements that need clearing up:

Why was the PCL-C chosen? There is a military version of this checklist.

The PCL-C was chosen for a variety of reasons; although this analysis is military centric, the participants are a part of a larger study that included civilians. As the PCL-M only assesses military trauma, which civilians will not have, and to account for a larger range of traumas that may cause PTSD (domestic violence, sexual assault, vehicle motor accidents etc.) the PCL-C was used.

Why was the PCL-C chosen and not the current PCL-5? The PCL-C is based on DSM-IV criteria, which have been outdated for almost 10 years. 

When the study began enrolment (2014) PLC-C was a relatively new tool, and although we agree that it is a strong tool built on the most current version of the DSM-V, multiple studies have shown that there is no significant difference between the PCL-5 and PCL-C (including in a military cohort, PMID: 33904913).

I’m even more confused, as I think you have not assessed trauma-exposure for the PCL-C, no? So, you only looked at symptoms, not trauma-exposure? Or is that done through the combat exposure scale? But, that’s odd, since the combat exposure scale would suggest that you used the PCL-M. So, what if someone has no combat exposure, but they might have had trauma-exposure in another way? In short: Has there been an selection on trauma-exposure? 

Since the PCL-C is not military specific (PCL-M) the combat exposure scale (CES) is commonly used to assess a service members military trauma exposure. Although this was statistically different in pairwise comparisons, when controlled for in the model all significant biomarkers retained their significance.

When was this study performed? 

This analysis is a part of a much larger study, for which nrollment began in 2014-May2019 (PLC-5 was just starting to emerge and protocol was written with PCL-C)

Line 142: “these are suggested cutoffs for military populations and have been previously used to assess PTSD symptom severity in SMVs [21].” The reference refers to the PCL-M, not the PCL-C. 

Correct. To remove confusion (and reduce the number of self-citations) we have removed that sentence and reference.

It isn’t clear is the alpha Cronbach scores refer to previous studies or the current studies. So: the alpha scores, are these for the currently used scales? 

We are not validating any tool in our analysis. The Cronbach scores are in reference to previous studies with validated Cronbach scores.

I don’t understand the population. An inactive reserve or a reserve component: do they see combat exposure? Wouldn’t they probably have not seen exposure? So, it might they don’t have PTSD, because they don’t fulfil criterion A. Which is quite important if the study is to look at relationships in explaining the role of a disorder. It might be that you are simply measuring stress symptoms. 

We apologize that this was unclear and we would like to address the reviewers concern and confusion about various military components. The authors are all extremely well versed in military service whether that comes in the form of teaching at USUHS or being a retired service member. The corresponding author (Hentig) is a retired service member who provides the following explanation: Our military was at a high deployment tempo from 2001-2021 in which millions of service members deployed, including most reserve units. Additionally, it is not uncommon for active-duty service members to transition from active duty to reserve or inactive reserve. The military component a service member is currently designated as is not a strong indicator of their deployment history or combat exposure.

The PHQ-9 is lower among people with PTSD than it is among people without PTSD? That’s very odd. PTSD and depression are highly correlated. The non-PTSD group has on average major depression. If this is not a mistake, I’m not sure in what way you can use these as a control group. The authors also bypass the large difference between the groups: one group has on average major depression disorder, the other does not. Even if the tests show that there is no statistical difference, it is an actual difference. Statistical test do not take into account diagnostic criteria. 

We agree that this was an odd finding, however, as the reviewer alluded to there is no statistical difference between PHQ-9 groups.  We did not collect any qualitative data from these patients, and as such we are limited to quantitative methods which suggest there is no difference. However, to address the reviewer’s concerns, we have placed PHQ-9 into the model to assess the potential impact on significance and all biomarkers retained their significance.

I think the authors should look more into how these groups can be used as control and PTSD group. Because I don’t think they can be compared, as they are quite similar. 

We are not attempting to compare a high PTSD group to a control group, but rather high PTSD to an ethnographically relevant population, in this case service members who undergo similar occupational stressors and traumas with self-reported low PTSD scores. The similarities between the populations is what allows for the impact of PTSD symptoms to associate with biomarkers that are known to play roles in a variety of physiological and psychological dysfunction.

So, I would suggest that the authors make a better case of why they can compare these groups. Also, if it is a problem that there sample has people who have combat exposure and others who on the reserve list. Why they use the PCL-C. Et cetera. 

We are not attempting to compare a high PTSD group to a control group, but rather high PTSD to an ethnographically relevant population, in this case service members who undergo similar occupational stressors with self-reported low PTSD scores. A service member with a designation of reservist or even active reservist is not indicative of their deployment history or previous combat exposure. The PCL-C has been found to have excellent agreement with the PCL-5 on individual items, PTSD diagnosis, and total score. 

Finally, I am unsure if the many self-references are necessary. 

We have reduced the number of self-references from 8 to 4, below the recommendation of the editor of 5.

Reviewer 2 Report

In the present study, the Authors aimed to assess whether there were differences in the peripheral blood concentrations of CRP, IL1RA, IL6, IL8, IL10, and 89 TNFα in a cohort of service members and veterans (SMVs) with a history of traumatic brain injury (TBI), with a lengthy average time since injury, who were either symptomatic or asymptomatic for post-traumatic stress disorder (PTSD). Although some of these cytokines have been previously reported, CRP, IL1RA, and IL8 have not been studied specifically in the context of SMVs with TBI, nor have they been linked to PTSD symptom severity. Furthermore, the Authors reported how these cytokines might correlate with measures of behavioral symptoms, including PTSD, depression, and post-concussive syndrome (PCS) symptoms, as well as health-related quality of life (HRQOL).

Overall, I found this study timely, original, well-conducted and scientifically sound: enjoyed reading it! However, I have some minor suggestions aimed at improving the quality of the paper, and these are outlined below:

1) In the introduction, a brief note on the clinical characteristics of PTSD that might be often a multidimensional disorder which includes some subtypes with different neurobiological underpinnings (i.e. prevalent flashbacks, prevalent avoidance, and so on) and its relationships with suicide risk, should be added with appropriate references (see dois: 10.1080/13651501.2019.1699575 and 10.2174/1389450116666150506114108).

2) Were the participants consecutive or randomly selected?

3) Was also the presence of an intellectual disability evaluated, how, and used as an exclusion criterion?

4) Nothing changes to the worth of the study, in my opinion, but I guess why the Authors decided to use for severity of depression symptoms the PHQ-9 instead of, for example, MADRS, which would permit the analysis of subscales or single items and are clinician-rated. Or why don't use both patient and clinician rating scales? Maybe it should be added to the limitations.

5) Was BMI objectively or subjectively assessed?

6) In tables, I suggest to report the full statistics rathen than the simple p value. Besides, in group comparisons, reporting effect sizes would be useful and strenghten the results.

7) In Figure 2, scatter plots of significant correlations in the PTSD high group, please report also the r value.

Author Response

We thank the reviewers for their time and effort. Reviewing is not always an easy task, but an essential task. Thank you for your comments, suggestions, and questions that ultimately have helped shape our manuscript into a better version.

Overall, I found this study timely, original, well-conducted and scientifically sound: enjoyed reading it! However, I have some minor suggestions aimed at improving the quality of the paper, and these are outlined below:

1) In the introduction, a brief note on the clinical characteristics of PTSD that might be often a multidimensional disorder which includes some subtypes with different neurobiological underpinnings (i.e. prevalent flashbacks, prevalent avoidance, and so on) and its relationships with suicide risk, should be added with appropriate references (see dois: 10.1080/13651501.2019.1699575 and 10.2174/1389450116666150506114108).

Thank you. We agree with the reviewer that including a brief description of associated subtypes, symptoms, and risks of PTSD was needed. To not over burden the intro the following was added:

Lines 48-51:” PTSD is a complex and multidimensional disorder which includes various subtypes, symptoms, and associated risks with different neurobiological underpinnings (e.g., intrusive thoughts, avoidance, anger, substance abuse, isolation) and has a strong relationship with suicide risk.”

2) Were the participants consecutive or randomly selected?

We are unsure what the reviewer is asking here. This was not a clinical trial and no groups were assigned. We had open enrollment of a larger study that involved longitudinal follow up. Only once the study was closed out, participants were then stratified based off from self-report symptoms.

3) Was also the presence of an intellectual disability evaluated, how, and used as an exclusion criterion?

The mental capacity to consent to the study was assessed and the NIH toolbox was utilized to assess cognitive ability. Otherwise, no, the presence of a specific intellectual disability was not evaluated. In the present study, all analyzed individuals were service members and as such presumed to meet the minimal requirements of health dictated by the DoD. The presence of an intellectual disability would be warrant for a medical discharge or rejection of service.

4) Nothing changes to the worth of the study, in my opinion, but I guess why the Authors decided to use for severity of depression symptoms the PHQ-9 instead of, for example, MADRS, which would permit the analysis of subscales or single items and are clinician-rated. Or why don't use both patient and clinician rating scales? Maybe it should be added to the limitations.

Dr. Roy, one of the senior authors was a part of the team that developed the PHQ-9

We appreciate this recommendation and will strongly consider alternative use in future studies. In the present study, the use of PHQ-9 is a routinely used and validated assessment within a military community.

We have added the following line acknowledging the limitation;

Lines 337-338: “Finally, although PHQ9 is a robust and routinely used tool, others exist that permit the analysis of subscales or single items and are clinician-rated.”

5) Was BMI objectively or subjectively assessed?

Yes, BMI was objectively assessed. Pairwise comparisons should a significant difference in BMI between groups, however when controlled for biomarkers with significance remained significant.

6) In tables, I suggest to report the full statistics rathen than the simple p value. Besides, in group comparisons, reporting effect sizes would be useful and strenghten the results.

We have added full statistics to the demo table, reported effect sizes for significant comparisons in line 259-261for all significant biomarkers, and full statistics for biomarkers can be found in the supplemental table.

7) In Figure 2, scatter plots of significant correlations in the PTSD high group, please report also the r value.

The r value, denoted as r, can be found at the upper right corner of each scatter plot.

Round 2

Reviewer 1 Report

I thank the authors for the thorough revision and answers.